# The Power of LC-MS Based Multiomics: Exploring Adipogenic Differentiation of Human Mesenchymal Stem/Stromal Cells

**DOI:** 10.3390/molecules24193615

**Published:** 2019-10-08

**Authors:** Evelyn Rampler, Dominik Egger, Harald Schoeny, Mate Rusz, Maria Pires Pacheco, Giada Marino, Cornelia Kasper, Thomas Naegele, Gunda Koellensperger

**Affiliations:** 1Institute of Analytical Chemistry, University of Vienna, Währinger Str. 38, 1090 Vienna, Austria; harald.schoeny@univie.ac.at (H.S.); mate.rusz@univie.ac.at (M.R.); gunda.koellensperger@univie.ac.at (G.K.); 2Vienna Metabolomics Center (VIME), University of Vienna, Althanstraße 14, 1090 Vienna, Austria; 3Chemistry Meets Microbiology, Althanstraße 14, 1090 Vienna, Austria; 4Institute for Cell and Tissue Culture Technologies, University of Natural Resources and Life Sciences, Muthgasse 18, 1190 Vienna, Austria; dominik.egger@boku.ac.at (D.E.); cornelia.kasper@boku.ac.at (C.K.); 5LMU München, Department Biology I, Großhaderner Str. 2-4, 82152 Planegg-Martinsried, Germany; Maria.Pacheco@biologie.uni-muenchen.de (M.P.P.); giada.marino@biologie.uni-muenchen.de (G.M.); Thomas.Naegele@biologie.uni-muenchen.de (T.N.)

**Keywords:** LC-MS, mesenchymal stem cells, stromal cells, fat differentiation, lipidomics, metabolomics, proteomics, multiomics, network analysis, mathematical modelling

## Abstract

The molecular study of fat cell development in the human body is essential for our understanding of obesity and related diseases. Mesenchymal stem/stromal cells (MSC) are the ideal source to study fat formation as they are the progenitors of adipocytes. In this work, we used human MSCs, received from surgery waste, and differentiated them into fat adipocytes. The combination of several layers of information coming from lipidomics, metabolomics and proteomics enabled network analysis of the biochemical pathways in adipogenesis. Simultaneous analysis of metabolites, lipids, and proteins in cell culture is challenging due to the compound’s chemical difference, so most studies involve separate analysis with unimolecular strategies. In this study, we employed a multimolecular approach using a two–phase extraction to monitor the crosstalk between lipid metabolism and protein-based signaling in a single sample (~10^5^ cells). We developed an innovative analytical workflow including standardization with in-house produced ^13^C isotopically labeled compounds, hyphenated high-end mass spectrometry (high-resolution Orbitrap MS), and chromatography (HILIC, RP) for simultaneous untargeted screening and targeted quantification. Metabolite and lipid concentrations ranged over three to four orders of magnitude and were detected down to the low fmol (absolute on column) level. Biological validation and data interpretation of the multiomics workflow was performed based on proteomics network reconstruction, metabolic modelling (MetaboAnalyst 4.0), and pathway analysis (OmicsNet). Comparing MSCs and adipocytes, we observed significant regulation of different metabolites and lipids such as triglycerides, gangliosides, and carnitine with 113 fully reprogrammed pathways. The observed changes are in accordance with literature findings dealing with adipogenic differentiation of MSC. These results are a proof of principle for the power of multimolecular extraction combined with orthogonal LC-MS assays and network construction. Considering the analytical and biological validation performed in this study, we conclude that the proposed multiomics workflow is ideally suited for comprehensive follow-up studies on adipogenesis and is fit for purpose for different applications with a high potential to understand the complex pathophysiology of diseases.

## 1. Introduction

In the context of obesity-related diseases it is essential to understand the molecular mechanisms that govern the formation of fat cells from stem cells. Mesenchymal stem/stromal cells (MSCs) are multipotent stem cells and the actual progenitors of fat cells/adipocytes [1]. MSCs are easily accessible as they can be derived from surgical waste or biopsies, i.e., bone marrow, birth-associated tissue, and adipose tissue. Due to their high proliferation capacity, immune-modulatory functions, and potential to differentiate into multiple lineages, MSCs carry a huge clinical promise in cell-based therapies, immunomodulation, and tissue engineering [2,3,4,5]. In contrast, induced pluripotent stem cells (iPSCs) are still not considered as safe since they are genetically not stable [6]. Furthermore, MSCs are ethically not questionable like embryonic stem cells. Today, adipose tissue is the predominant cell source for MSCs [2,7] as they can be obtained more easily and efficiently than from bone marrow [4]. The process of adipocyte formation, also known as adipogenesis, is regulated by a network of complex molecular processes. Adipocyte differentiation acts via signal transduction of hormones, growth factors, metabolites, lipids, and specific protein receptors mediating external growth and differentiation signals through a cascade of intracellular events. In general, fat cell differentiation is triggered by the ceramide and peroxisomal proliferator-activated receptor gamma (PPARγ) [8,9,10]. Insulin and insulin-like growth factor 1 (IGF-1) are required for fat cell formation and activate distinct downstream signal transduction pathways such as (1) a phosphorylation-dephosphorylation mechanism subsequent to IGF-1 tyrosine phosphorylation or (2) insulin activation of the phosphatidylinositol 3-kinase pathway [11]. During the terminal differentiation phase, adipocytes’ de novo lipogenesis and insulin sensitivity is highly increased. Protein and mRNA levels for enzymes (e.g., stearoyl-CoA desaturase, glycerol-3-phosphate dehydrogenase, and fatty acid synthase) involved in triacylglycerol metabolism are increased 10–100-fold upon fat cell formation [11].

In order to follow MSC differentiation into fat cells, suitable analytical workflows are necessary. Omics-based strategies are ideal for adipocyte differentiation analysis as they are based on the observation of a subset of molecules in one organism. In this work, we apply a multiomics workflow combining metabolomics, lipidomics, and proteomics. Mass spectrometry (MS) based multiomics strategies are extremely powerful due to their unrivaled potential to monitor an entire set of molecules in order to follow cellular metabolism and perturbations [12,13,14]. The combination of liquid chromatography (LC) with high-resolution MS is ideally suited for multimolecular chemical analysis, as LC-MS enables identification and quantification by (1) retention time, (2) accurate mass, and (3) fragmentation pattern of hundreds to thousands of compounds in parallel. Metabolomics of stem and fat cells provides information on the phenotypic change of small metabolites involved in the differentiation process. With lipid droplet formation as a hallmark of adipogenic differentiation [11], lipidomics is able to reflect elevated lipid content at the same time providing information on MSCs and adipose tissue lipid patterns. Proteomics on the other hand, enables to monitor important protein key players involved in fat cell formation. In our workflow, we aim to merge targeted quantification and non-targeted screening of metabolites and lipids to follow MSC differentiation in adipocytes. The absolutely accurate quantification of several hundred lipids remains a field of analytical development, as desirable lipid compound-specific standardization is a challenge due to the complexity of the lipidome. Recently, we developed the novel Lipidome Isotope Labeling of Yeast (LILY) technology for in vivo labeling of yeast to produce eukaryotic lipid standards [15,16]. With our in vivo labeling approach, several hundred stable-isotope-labeled lipids can be simultaneously produced, enabling us to follow lipids as key players in fat cell differentiation. Isotope dilution approaches involving stable-isotope-labeled lipid and metabolite standards can account for any error accumulated in the analytical chain of sample preparation, storage, and measurement. Using isotope dilution in combination with high-end chromatography and MS instrumentation for the analysis of MSCs and adipocyte differentiation will provide us with enhanced data matrices as an ideal starting point to build metabolic models. MSC differentiation affects numerous cellular metabolic pathways and signaling cascades. Mathematical metabolic models can reveal metabolic characteristics in these complex regulatory networks [17]. Based on such genome-scale models, essential metabolites, lipids, and proteins which are key to cell differentiation can be identified. For example, a genome-scale study on metabolic network model of bone marrow-derived MSCs has previously been presented [18]. A successful model was developed to identify ways to increase stem cell proliferation and differentiation, providing evidence for the key role of mathematical modelling in biomedical research. Thus, applying a combination of genome-scale modelling and experimental large-scale omics-analysis together with multivariate statistics or pattern recognition [19] promises to yield significant advances in the complex field of adipogenic differentiation. 

The mechanistic understanding of fat cell development in the human body is essential for the treatment of obesity-related diseases. However, a high fraction of studies on the mechanisms that direct adipogenic differentiation rely on mouse models or murine cell lines [3,20,21,22,23]. In the context of cell-based therapies, translational aspects are of utmost importance as cells from different species might act differently. For example, the in vitro proliferation of human MSCs exhibits a relatively low frequency of oncogenic transformation in stark contrast to murine MSCs which frequently gain chromosomal defects in vitro and often produce fibrosarcomas when injected back into mice [5,24,25]. To enable potential translation of our results, we decided to use human primary MSCs in early passages for this study. In this work, we present a novel MS-based multiomics strategy to analyze adipogenesis in human MSCs relying on the combination of high-end MS and chromatography, isotopic dilution and network analysis.

## 2. Results

### 2.1. Development of a Multiomics Workflow for the Analysis of Adipogenic Differentiation from MSCs

Primary adipose tissue-derived MSCs (MSC0) were cultivated in adipogenic differentiation medium for 21 days (MSC21). Successful adipogenic differentiation was observed by morphological changes of the cells and lipid vacuole formation (Figure 1). In order to perform a representative systems biology experiment, the multiomics data should ideally be generated from the same set of samples to allow for direct comparison under the same conditions [26]. To meet these requirements, we applied an adapted two-phase extraction procedure for MSC and fat cells (n = 5) to sequentially extract lipids, metabolites, and proteins [10]. Sequential lipid, metabolite, and protein extraction by MTBE provided the advantage of multimolecular information from one sample without the drawback in sample handling by protein precipitation in the interphase, then chloroform/methanol/water extraction systems were used [27]. We modified the original SIMPLEX protocol [10] by using nitrogen quenching so that trypsin was avoided and less amounts of human MSCs and adipocytes (~10^5^ cells) were needed. We developed a multiomics strategy based on hyphenated high-end mass spectrometry and chromatography for non-targeted screening and targeted quantification.

Protein information was generated by state of the art LC-MS/MS shotgun proteomics and was used to build metabolic networks linked to human genomic sequences. Metabolite and lipid profiling in MSCs and adipocytes was performed using a novel HILIC-RP dual injection strategy combined with high-resolution Orbitrap-MS [28]. Merging metabolomics and lipidomics assays is usually compromised by different extraction procedures needed (high polarity ranges from very polar small metabolites to medium-mass nonpolar lipids), and typically the two approaches deliver separate data sets. Using the parallel HILIC-RP method, two extracts can be measured on two columns in one analytical run so that metabolite and lipid information is generated in 32 min. Concentrations ranging over three to four orders of magnitude can be assessed by the HILIC-RP-MS assay [28]. The addition of ^13^C-labeled metabolite/lipid standards enabled us to merge targeted quantification with untargeted compound screening in MSCs and adipocytes. Low sample amounts of MSCs and adipocytes (~10^5^ cells) were needed for detecting fmol absolute on the column in the parallel HILIC-RP method. This is significantly lower than a recent shotgun lipidomics approach used for the quantitative analysis of white and brown adipose tissue, where pmol of lipids were theoretically detectable [29] highlighting the sensitivity advantage when LC-MS based workflows are applied.

### 2.2. Targeted Lipidomics and Metabolomics Comparing MSCs and Adipocytes

For targeted analysis, we employed isotopically ^13^C-labeled metabolite and lipid standards produced in our laboratory for standardization [15,30,31] to follow fat cell differentiation of MSCs. Absolute compound-specific quantification (n = 3) of 12 lipids and 54 metabolites (compound- specific quantification using endogenous/^13^C-labeled pairs) was performed using external calibration (8 standards within the calibration range of 0.1–25 µM), with internal standardization by in-house produced ^13^C-labeled standards and protein normalization. The selection of the targeted analytes was performed on the available standards mixes and corresponding ^13^C-labeled lipids/metabolites (Appendix A). All lipids and metabolites quantified in MSCs/adipocytes were in the range of nmol to fmol range per mg protein with limits of detection of fmol absolute on column. Isotope dilution and the use of the LILY lipids enhanced the quantification significantly (Figure A1 and Figure A2), i.e., triglycerides (TG) 54:6 (R^2^ of 0.9812 vs. 0.7146 for label-free quantification). Once more showing the advantage of using our in-house produced standards for quantitation [15,16]. The comparison of the targeted metabolite and lipid quantification in MSCs and adipocytes revealed significant downregulation (fold change > 2, *p*-value < 0.05) of GMP, S-adensoyl-methionine, asparagine, L-cystathionine, adenosine, 5’-deoxy-5’-methylthioadenosine, and adenine in adipocytes. The amino acids showed a general trend to be less abundant in adipocytes (Figure A3A), whereas triglycerides and lipid precursors such as carnitine and propionyl-L-carnitine were increased upon differentiation (Figure A3B,C), which can be explained by the formation of lipid droplets upon differentiation of the MSCs. Separation of MSC and adipocytes was observed by principal component analysis using the targeted metabolite and lipid data set (Figure 2A,B). Due to the low number of replicates (n = 3) a profound statistical evaluation of significance is not possible here. However, adipocyte (MSC21) metabolite and lipid data showed a much higher variance than before differentiation (MSC0). This indicates a diversification of the metabolome and lipidome due to differentiation which remains to be tested for its significance in future studies.

### 2.3. Non-targeted Lipidomics and Metabolomics Comparing MSCs and Adipocytes 

Non-targeted lipid and metabolite analysis revealed significant changes (fold change > 2, *p*-value < 0.05,) comparing undifferentiated MSCs (MSC0, cells before differentiation initiation, n = 5) with MSCs differentiated into the adipogenic lineage (MSC21, cells after 21 days of adipogenic incubation, n = 5). A general increase in molecular complexity in differentiated adipocytes was observed indicated by the higher number of identified metabolites and lipids (Figure 3). After filtering (Compound Discoverer 3.0: background removal using extraction and eluent blanks, area > 50,000, MetaboAnalyst: Interquartile range) and normalization (mean-centered and divided by the standard deviation of each variable in MetaboAnalyst), 171 compounds were significantly upregulated and 24 compounds were downregulated in adipocytes. Overall, 2650 compounds were identified in the lipid fraction of both sample groups which is three times higher compared to the metabolite fraction with 839 compounds (Figure 3). This observation reflects the formation of lipid droplets as a hallmark for adipogenic differentiation, and is consistent with a general trend of summed area increase in adipocytes detected for most of the abundant lipid classes, namely triglycerides (TG), phosphatidylcholine (PC), lysophosphatidylcholine (LPC), sphingomyelin (SM), ceramides (Cer), and diglycerides (DG) (Figure A4, Appendix A). The upregulation was significant in TGs (Figure A4), which are the main constituents of lipid droplets [3]. For the other lipid classes detected belonging to the category of phospholipids, no significant changes (phosphatidylethanolamines (PE), phosphatidylinositols (PI), phosphatidylserines (PS), lysophosphatidyl-ethanolamines (LPE)) were detected. Overall, an indication towards phospholipid remodeling (Figure A4, Appendix A) upon adipogenic differentiation/lipid droplet formation was observed [33].

### 2.4. Biological Validation and Data Interpretation of the Multiomics Workflow

#### 2.4.1. Proteomics Network Reconstruction, Metabolic Modelling and Pathway Analysis

The shotgun proteomic analysis of non-differentiated and differentiated MSCs identified 1869 proteins (Appendix A, protein IDs, n = 2) and revealed significant dynamics of the cellular proteome during the differentiation process. The observed protein changes are in accordance with previously published proteomics data on human adult stem cell adipogenesis, e.g. matching upregulation of apolipoprotein, fatty acid binding protein, glycerol-3-phosphate dehydrogenase, fatty acid synthase, phosphoglycerate synthase, and downregulation of platelet activating factor, acetylhydrolase 1B, and annexin was observed [34]. The upregulation of glycerol-3-phosphate dehydrogenase and fatty acid synthase was also reported as a consequence of increased triacylglycerol metabolism upon lipid droplet formation in adipocytes [11].

Based on the human genome-scale consensus model Recon 2.4, proteomic data were applied to reconstruct context-specific networks which reflect the specific proteomic constitution of MSCs before and after differentiation. In total, 165 metabolic reactions and processes were fully reprogrammed on the proteome level during differentiation (Figure A5). The strongest effects were observed for reprogramming of reactions in fatty acid metabolism (oxidation, synthesis, eicosanoid, inositol phosphate, and bile acid metabolism) and transport processes (peroxisomal and extracellular, as well as endoplasmatic reticular transport) (Appendix A, Figure A5).

Using the protein information together with the generated metabolite and lipid (combining targeted and non-targeted lists with median areas/concentrations for comparison of MSCS and adipocytes) data (Appendix A), multiomics pathway analysis was performed (OmicsNet [35]). Comparing MSCs and adipocytes, 113 significantly regulated pathways (*p*-value > 0.05) (Appendix A) were identified in four different correlation networks including one major network (continent network-101 regulated pathways) and three subnetworks (island networks for glycerolipids-7 pathways; glutathione disulfide-2 pathways, carnitine- 3 pathways). Adenosine triphosphate (ATP) and adenosine monophosphate (AMP) showed the highest correlation degree (922, 141 respectively) within the protein-metabolite network (Figure 4A,C) due to the expected change in energy metabolism and phosphorylation-dephosphorylation upon fat differentiation [11]. Several biological relevant and expected active pathways were identified (Figure 4A–C) including (1) glycerophospholipid signaling (Figure 4A,B, Appendix A, network 1) [10,36], (2) the insulin pathway (Figure 4C, Appendix A, network 1) [8,9,10], (3) glycerolipid (triglycerides regulating) pathway with corresponding fat digestion and absorption (Figure A6A, Appendix A, subnetwork 2) [11,37,38], (4) carnitine regulation within the fatty acid metabolism, PPAR signaling, and adipocytokine signaling pathways (Figure A6B, Appendix A, subnetwork 4) [39,40,41], and (5) glutathione disulfide within arachidonic acid metabolism and oxidation related pathways (Appendix A, subnetwork 3) [42,43,44]. Interestingly, triglycerides- and carnitine-related pathways were separated as individual subnetworks (Figure A6A,B, Appendix A). Both compound classes are known to be highly regulated [11,39] in adipogenesis and will be discussed in more detail below.

#### 2.4.2. Carnitine Regulation

Increased concentrations of carnitine and propionyl-L-carnitine were observed in adipocytes by targeted analysis (Figure A3C) with a significant regulation of carnitine related pathways (Figure A6B, Appendix A, subnetwork 4). Carnitine synthesis is necessary for fatty acid oxidation and fatty acid transport from the cytosol to the mitochondria [45] and is known to improve insulin sensitivity of adipocytes [46], an inherent characteristic of adipose tissue. Interestingly, upregulated acyl-carnitine levels are associated with the promotion of the differentiation process of embryonic stem cells in different cell types [42]. Additionally, carnitine and acetyl-carnitine were found to reduce adipogenesis but stimulate osteogenesis and chondrogenesis in human MSCs by regulating the mitochondrial metabolisms [39]. It is also known to attenuate lipid accumulation in adipocytes via downregulation of PPARγ, a major adipocyte-specific transcription factor [47].

#### 2.4.3. Trigylcerides Upregulation in Lipid Droplets of Adipocytes

A major increase of TGs in adipocytes was observed in the targeted lipid analysis (Figure A3B) and the number of identifications received by non-targeted lipidomics (Figure 3). Additionally, triglycerides were involved in several significantly regulated pathways of the multiomics data (Figure A6A, Appendix A, subnetwork 2). Comparing the 10 most regulated compounds in the classes of MSCs and adipocytes (from the combined untargeted metabolite and lipid data set), several interesting *m*/*z* variables were found by partial least square and heat map analysis (MetaboAnalyst 4.0 [32]). Some of them could be further identified by a database search (Lipid Search) and manual curation. Four upregulated triglycerides in adipocytes were annotated based on accurate mass, fragmentation and retention time, namely TG (16:0_16_1_18:0), TG (16:0_16:1_18:1), TG (16:1_18_0_18_0), and TG (16:0_18:1_18:1). TGs are the major constituents of lipid droplets representative of successful adipogenic differentiation. Expected lipid droplet formation was induced in the adipocytes (Figure 1) upon medium change and reflected in the list of regulated proteins by the increase of, for example, fatty acid synthase, glycerol-3-phosphate dehydrogenase, and malic enzyme, which are all known to increase the triglyceride metabolism [11].

#### 2.4.4. Gangliosides are Involved in Adipocyte Differentiation and Insulin Sensitivity

The *m*/*z* for the gangliosides GM3(d18:1/16:0) and GM3(d18:1/22:1) were listed within the 10 most regulated compounds identified (partial least square, heat map, MetaboAnalyst [32]) when comparing MSCs and adipocytes. We annotated them manually by *m*/*z*, fragmentation, and matching retention times (in positive and negative MS mode), and observed an upregulation in adipocytes by a factor of 20 and 30 respectively. Comparing the ganglioside glycosphingolipid fragmentation pattern (Figure A7), the ceramide moiety was determined in the positive ion mode (characteristic sphingosine fragments at *m*/*z* 264 and 282) and the glycan residue was annotated in the negative ion mode (characteristic N-acetylneuraminic acid fragment ion at *m*/*z* 290). Additionally, a series of gangliosides (GM3(32:1;2) GM3(32:0;2), GM3(33:1;2), GM3(34:0;2), GM3(34:2;2), GM3(35:1;2), GM3(36:0;2), GM3(36:1;2), GM3(36:2;2), GM3(38:1;2), GM3(40:0;2), GM3(40:1;2), GM3(40:2;2), GM3(41:0;2), GM3(41:1;2), GM3(41:2;2), GM3(42:0;2), GM3(42:1;2), GM3(43:1;2), and GM3(43:2;2)) was identified by *m/z*, elution order depending on the number of double bonds and fatty acid chain length, as well as matching retention time (elution: 22.3–28.3 min) between the different MS modes (+/−). Ten (GM3(33:1;2), GM3(35:1;2), GM3(36:0;2), GM3(41:0;2), GM3(41:1;2), GM3(41:2;2), GM3(42:0;2), GM3(42:1;2), GM3(43:1;2), and GM3(43:2;2)) of the 22 identified gangliosides were only found in adipocytes and all other gangliosides were found in lower abundances in MSCs. This is of special interest, as GM3 gangliosides are part of the sphingolipid metabolism identified by protein-metabolite network data (Appendix A). The connection of ganglioside increase upon adipogenesis was further supported by the upregulation of the ganglioside GM2 activator protein in adipocytes, which is able to extract GM2 molecules from membranes to hydrolyze them to GM3. The enzymatic activity of ganglioside GM2 activator protein strongly depends on the sphingolipid and membrane lipid composition [48]. Gangliosides were proposed as a new class of stem cell markers, e.g. GD1a increase for favored osteogenesis in MSCs [49]. They were further found to be involved in the differentiation of MSCs into osteoblasts and neural cells [50] and are generally associated with the insulin metabolic signaling in adipocytes [51]. Insulin is required for fat cell formation activating distinct signal transduction pathways, such as phosphorylation-dephosphorylation or the phosphatidylinositol-3-kinase pathway [10]. Major reprogramming of related pathways (insulin, glycerolipid, sphingolipid, phosphatidylinositol pathway) was also confirmed by the multiomics data (Figure 4, Appendix A, subnetwork 1). Additionally, we observed upregulation of perilipin-4 upon adipocyte differentiation of MSCs (Appendix A) which is an adipocyte phosphoprotein present in the periphery of lipid storage droplets [3,11].

## 3. Discussion

The presented multiomics study on the adipogenic differentiation of MSCs is a proof of principle for the power of multimolecular extraction procedures combined with orthogonal LC-MS assays and network construction. Merging metabolomics and lipidomics assays was possible by adapting (no trypsin, less starting cell number) the original SIMPLEX protocol [10] for sequential lipid, metabolite, and protein extraction. Followed by the HILIC-RP dual injection setup, metabolite and lipid information was generated from the same sample in one analytical run. Additionally, targeted quantification by ^13^C-labeled metabolite/lipid standards could be performed with untargeted compound screening in the same assay, so that sample amounts needed were significantly reduced. The ^13^C-labeled metabolites/lipids can be potentially used for normalization of non-targeted LC-MS based multiomics data, which was recently shown to be possible for lipidomics data [52] and represents an important future goal. LC-MS assays are especially useful when low abundant metabolites and lipids are of interest as the chromatographic separation offers cleanup of the sample matrix, higher linear dynamic range, and additional compound identification by retention time information. The power for the detection of unknown low abundant compounds was impressively shown by the example of GM3 gangliosides in MSCs and adipocytes. None of the gangliosides could be annotated by a database search, although the used in silico database included all types of gangliosides (GM1, GM2, GM3). Only when comparing the non-targeted datasets of MSCs and adipocytes, two of these high mass (>1000 *m*/*z*) glycosphingolipids showed up as interesting differentiating unknown compound. Using manual curation of the fragmentation pattern, matching retention times in the positive and negative ion mode, and retention time order, 22 different gangliosides could be successfully annotated and were further confirmed by the upregulation of the ganglioside GM2 activator protein. Despite the limited sample number of the assay, all biological observations for the regulation of triglycerides [3,8], gangliosides [51], and reprogrammed pathways [23] in adipocytes are in accordance with the literature. Further biological interpretation of the data is beyond the scope of this work. Future studies will focus on in-depth analysis of fat differentiation involving time-series experiments, tracer studies, and the comparison of different fat cell tissue types. Additionally, targeted LC-MS based assays will be developed for metabolite and lipid classes relevant in adipogenesis such as triglycerides, acyl-carnitines, or gangliosides.

The presented biotechnology workflow based on adult human MSCs from surgical waste represents a general strategy to avoid animal experiments and induced pluripotent stem cells [6,53] for certain clinical studies and drug testing. The performed biological validation (literature search, identification of relevant regulated compounds and pathways) combined with the analytical method validation (analytical figures of merit; use of commercially available standards as well as in-house produced labeled metabolites and lipids; database annotation based on RT, accurate mass and fragmentation pattern; additional manual curation), lets us conclude that the presented workflow is fit for purpose for other cell culture or tissue-based multiomics studies. LC-MS based multiomics workflows enable to follow the complex pathophysiology of diseases among various other applications. Our multiomics workflow offers comprehensive multimolecular information based on a single sample (using the sequential extraction approach) and low analysis time. We hypothesize that reproducible diagnostic high-throughput LC-MS based assays can be implemented by automatic sample extraction and data evaluation.

## 4. Materials and Methods 

### 4.1. Cell Culture

The use of human tissue was approved by the ethics committee of the Medical University Vienna, Austria (EK Nr. 957/2011, 30 January 2013) and the donor gave written consent. Human adipose tissue-derived MSCs were isolated from a female donor (age 29) within 3–6 h after surgery as described before [54]. Briefly, adipose tissue was obtained from abdominoplasty, minced with scissors, and digested with collagenase type I (Sigma Aldrich, St. Louis, MO, USA). After several centrifugation and washing steps, the stromal vascular fracture was released in a cell culture flask (Sarstedt, Nümbrecht, Germany) with standard cell culture medium composed of MEM alpha (Thermo Fisher Scientific, Waltham, MA, USA), 0.5% gentamycin (Lonza, Basel, Switzerland), 2.5% human platelet lysate (hPL; PL BioScience, Aachen, Germany), and 1 U/mL heparin (Ratiopharm, Ulm, Germany), and incubated in a humidified incubator at 37 °C and 5% CO_2_. Subsequently, MSCs were selected by plastic adherence and cryo-preserved in 77.5% MEM alpha, 12.5% hPL, 10% DMSO (Sigma Aldrich), and 1 U/mL heparin (defined as passage 0, P0). For the adipogenic differentiation, cells were thawed at passage 0 and further expanded for two passages in T-flasks. After harvest via accutase (GE healthcare, Little Chalfont, UK) treatment, MSCs were seeded at 4.000 cells/cm^2^ in a 6-well plate (Sarstedt, Nümbrecht, Germany), coated with 2 µg/cm^2^ fibronectin (Sigma, St. Louis, MO, USA), and 2 mL of CM was added. Upon confluency, the medium was changed to adipogenic differentiation medium (ADM; Miltenyi, Bergisch-Gladbach, Germany), supplemented with 0.5% gentamycin, and the cells were cultured for 21 days, while the medium was changed every 2–3 days. Undifferentiated MSCs at day 0 served as a control. Subsequently, the cells were fixed with 4% paraformaldehyde (Sigma, St. Louis, MO, USA), washed with ddH_2_O and stained with Oil Red O solution (Sigma, St. Louis, MO, USA). The staining of lipid vacuoles was observed by bright-field microscopy (DMIL LED with camera ICC50HD; both Leica, Wetzlar, Germany).

### 4.2. Multiomics

All solvents were LC-MS grade. Metabolite and lipid standards were from Sigma, Carbosynth (Compton Berkshire, England) or Avanti Polar Lipids, Inc. (Alabaster, Alabama, USA) and were weighed and dissolved in an appropriate solvent. A multi-metabolite mix (148 metabolites) and multi-lipid mix (76 lipids) were prepared (a comprehensive list of the standards can be found in Appendix A). A metabolite and lipid ^13^C-labeled standard mix was prepared using the fully ^13^C-labeled internal standard from ISOtopic solutions e.U. (Vienna, Austria), combined with an in-house produced ^13^C-labeled lipid LILY triglyceride fraction and reconstituted in 50% MeOH. The MSC and adipocytes cells were quenched directly on the 6-well plate using liquid nitrogen prior to storage at −80 °C until sample preparation. The adherent cells (~10^5^ per sample) were harvested (5 samples per MSC0/MSC21 condition, 2 medium blanks) using a cell scraper and extracted with a mixture of methanol, methyl-tert-butylether, and 10 mM ammonium formate. The adapted SIMPLEX two-phase extraction preparation [10] enabled the collection of the lipid fraction (upper phase), the metabolite fraction (lower phase), and the protein pellet. All samples (n = 5 per condition) were used for non-targeted analysis. Part of the samples were spiked (n = 3 per condition) with ^13^C isotopically labeled metabolites and LILY lipids for targeted quantification. This fraction was used for absolute compound-specific quantification by external calibration (0.1, 0.25, 0.5, 0.75, 1, 5, 10, 25 µM) with the commercially available standards (multi-metabolite, multi-lipid mix) and internal standardization by ^13^C-labeled metabolites and lipids. 

A novel HILIC-RP DUAL injection strategy combined with high-resolution Orbitrap-MS was applied for the simultaneous analysis of small polar metabolites and lipids [28] of the MSC samples. Briefly, a Vanquish Duo UHPLC system (Thermo Fisher Scientific) equipped with an autosampler with two injection units, two binary pumps and a column compartment was used to enable parallel HILIC and RP analyses. A SeQuant^®^ ZIC^®^-pHILIC column (150 × 2.1 mm, 5 μm, polymer, Merck-Millipore) was used with gradient elution under alkaline conditions (mobile phase A: 90% 10 mM ammonium bicarbonate, pH 9.2/10% ACN; mobile phase B 100% ACN) combined with an Acquity HSS T3 (2.1 mm × 150 mm, 1.8 μm, Waters) RP column (mobile phase A: ACN/H2O 3:2, *v*/*v*, solvent B: IPA/ACN 9:1, *v*/*v*) using a VanGuard precolumn (2.1 mm × 5 mm, 100 Å, 1.8 μm). Two different gradients were applied for HILIC and RP with a total run time of 32 min as described before [28] using a flow rate of 300/250 µL min^−1^ and an injection volume of 5 µL. An additional 2-positional 6-port valve and the use of a T-piece in front of the mass spectrometer enabled to deliver the HILIC (0–11 min) or RP (11–32 min) eluent to the MS, while the other separation eluent was directed to waste. High-resolution mass spectrometry in positive and negative mode was conducted on a high field Thermo Scientific™ Q Exactive HF™ quadrupole-Orbitrap mass spectrometer equipped with an electrospray source using Full-MS (resolution 120,000 for quantification, positive mode only) and ddMS (resolution 30,000, positive and negative mode, Top-15) for identification. Detailed information on the dual injection HILIC-RP-MS workflow can be found in the supplementary information of Appendix C.

The protein fractions containing the isotopically spiked samples (n = 3) were determined by Bradford assay (Pierce™ Coomassie Plus Assay Kit) and used for normalization. The other protein pellets (n = 2) were used for shotgun proteomics including protein extraction, digestion and LC-MS/MS analysis. Proteins were further purified in a 6 M guanidine chloride and 10mM HEPES-KOH buffer (pH 7.5) containing protease inhibitors before being followed by precipitation in a methanol/chloroform/water (4:3:2 *v*/*v*/*v*) mixture. Precipitated proteins were resuspended in 6 M Urea, 2 M thiourea, and 10 mM HEPES-KOH (pH 7.5). For reduction and alkylation, DTT (10 mM) and iodoacetamide (50 mM) was added. Proteins were trypsin-digested overnight, and 1 µg was loaded on the column for LC-MS/MS analysis. LC-MS/MS analysis was performed on a nano-LC-system (Ultimate 3000 RSLC; Thermo Fisher Scientific) coupled to an Impact II high resolution quadrupole time-of-flight (Bruker) using a Captive Spray nano electrospray ionization source (Bruker Daltonics). The nano-LC system was equipped with an Acclaim Pepmap nanotrap column (C18, 100 Å, 700 µm 2 cm; Thermo Fisher Scientific) and an Acclaim Pepmap RSLC analytical column (C18, 100 Å, 75 µm × 50 cm; Thermo Fisher Scientific). The peptide mixture was fractionated by applying a linear gradient of 5% to 45% acetonitrile at a flow rate of 250 nL min^−1^ over a period of 80 min. The column temperature was set to 50°C. MS1 spectra were acquired at 3 Hz with a mass range from *m*/*z* 200–2000, with the Top-15 most intense peaks selected for MS/MS analysis using an intensity-dependent spectra acquisition time between 4 and 16 Hz. Dynamic exclusion duration was 0.5 min.

### 4.3. Data Analysis

Data analysis was performed by dedicated software for targeted (Skyline 4.2) and untargeted analysis (metabolites: Compound Discoverer 3.0; lipids: Lipid Search 4.1, Compound Discoverer 3.0; proteins: MaxQuant 1.6.2.10). For targeted analysis, normalization by ^13^C-labeled lipids/metabolites and protein content was performed. For non-targeted data analysis, background correction (by medium/blank samples) and normalization (mean-centered and divided by standard deviation in MetaboAnalyst) was performed prior to further statistical and pathway analysis using MetaboAnalyst 4.0 [32]. Adipocyte samples had a higher variation than MSC cells, which was considered for statistical analysis. Context-specific metabolic networks were reconstructed from shotgun proteomics data applying the FASTCORE algorithm [55] using a consistent version of Recon 2.4 [56]. FASTCC [55] was applied with the default flux activity threshold of 1e-4. Proteomic data was mapped to the consistent version of Recon2.4 via the Gene Protein Association rules (GPR rules) to obtain a set of core reactions, i.e., reactions that were found both in the proteomic data and the network model of Recon2.4. FASTCORE were used to identify the missing reactions necessary to obtain a consistent context-specific subnetwork for all differentiation states. OmicsNet was used for multiomics data integration, visualization and pathway analysis [35].

## Figures and Tables

**Figure 1 molecules-24-03615-f001:**
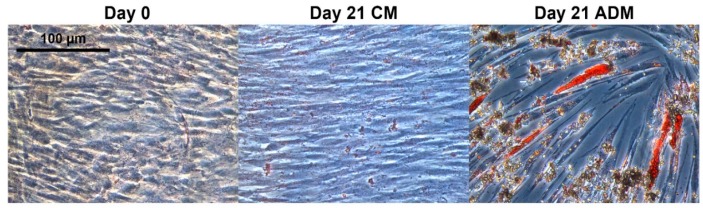
Oil Red O staining of adipose tissue-derived mesenchymal stem/stromal cells (MSCs) after 0 and 21 days cultivation in the adipogenic differentiation medium (ADM) or control medium (CM). The scale bar indicates 100 µm.

**Figure 2 molecules-24-03615-f002:**
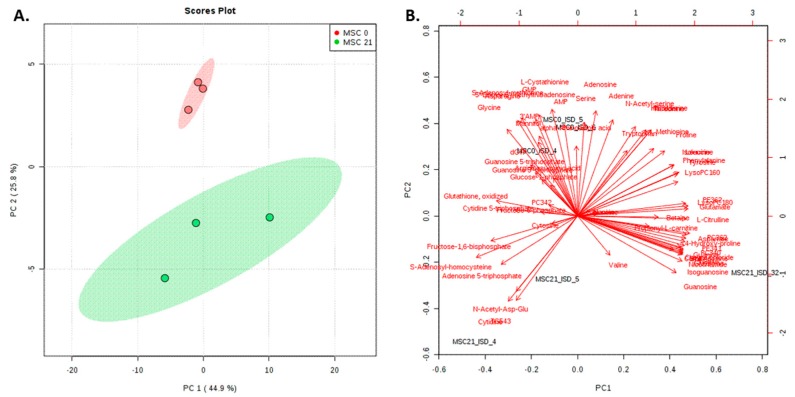
Principal component analysis (PCA) for the targeted lipid and metabolite analysis comparing MSCs and adipocytes. All compounds were quantified (54 metabolites, 12 lipids) using compound-specific quantification with ^13^C-labeled yeast extract. (**A**) PCA score plots in MetaboAnalyst 4.0 [32] explaining 71% of the variance by PC 1 and PC 2 showing separation of the MSCs (MSC0) and adipocytes (MSC21) despite the higher variation in adipocytes. (**B**) Biplot (MetaboAnalyst) including the loading information of the individual metabolites/lipids.

**Figure 3 molecules-24-03615-f003:**
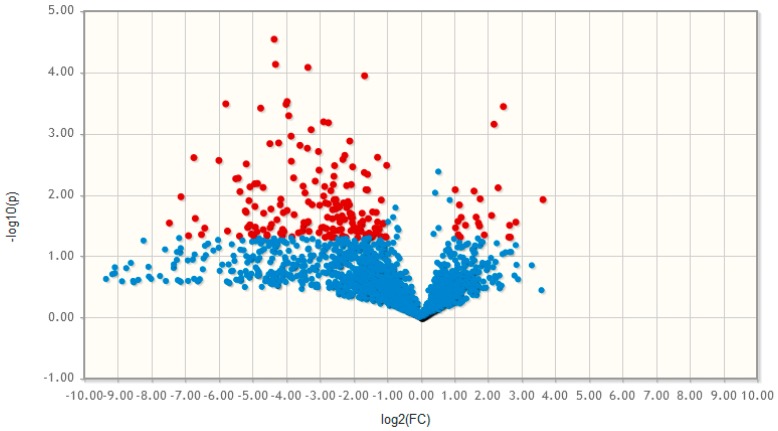
Differential analysis of lipids and metabolites. Volcano plot (derived from untargeted LC-MS) of compounds from primary MSCs and adipocytes using MetaboAnalyst 4.0 (with *p*-value settings 0.05, interquartile filter) [32]. HILIC-RP-HRMS (positive mode) revealed that 195 out of 3498 compounds were significantly regulated (*p*-value < 0.05, fold change (FC) > 2). Both lipids and metabolites were significantly upregulated in adipocytes compared to MSCs.

**Figure 4 molecules-24-03615-f004:**
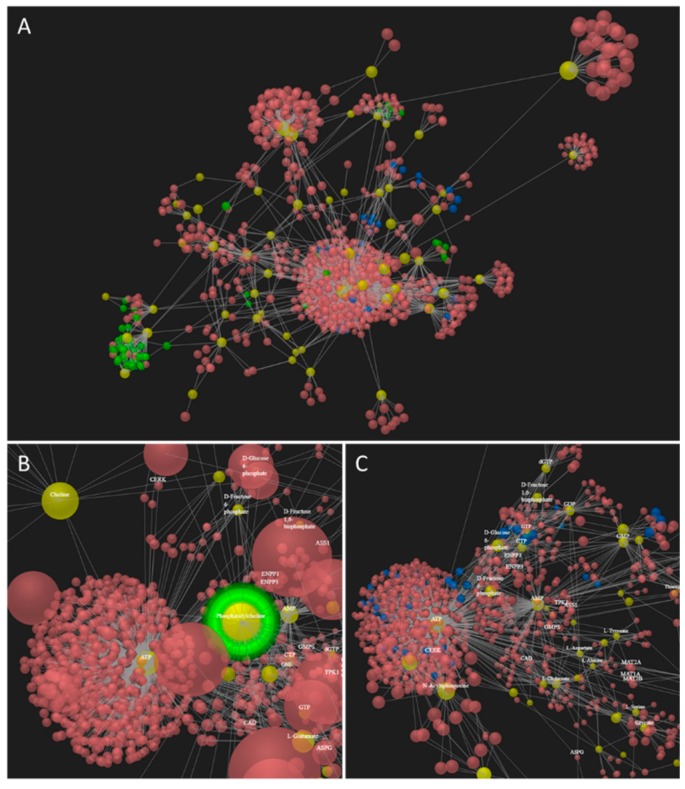
Multiomics network analysis of MSCs and adipocytes. Network analysis by OmicsNet [35] enabled to combine protein, metabolite, and lipid information, and identify regulated pathways in adipogenesis. (**A**) The network including most protein-metabolite interactions. Proteins are labeled in red and metabolites in yellow. The glycerophospholipid metabolism is labeled in green and the insulin pathway in blue. (**B**) Phosphatidylcholine as regulated lipid within the glycerophospholipid metabolism. (**C**) ATP and AMP as major regulated energy metabolites, additionally the insulin pathway is significantly regulated and labeled in blue.

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
