# Peer review of "The Power of LC-MS Based Multiomics: Exploring Adipogenic Differentiation of Human Mesenchymal Stem/Stromal Cells"

_molecules, 2019, doi:10.3390/molecules24193615_

Round 1

Reviewer 1 Report

The paper is well written even if it is someway quite difficult to read. Authors have to keep in mind that also readers, which are not experts of omics methods, could be interested in reading the paper. For this reason, they should make an effort to render their paper more understandable for general audience, this should be done without loosing the deepness of their analysis.

The proteomic data they present are related to differentiation into adipocytes of adipose derived mesenchymal stromal cells. It could be of interest if authors compare their data with available proteomics data related either to protein content or secretome of adipose  mesenchymal stromal cells. 

For example, see the paper published on Aging (Albany NY). 2016 Jul;8(7):1316-29. doi: 10.18632/aging.100971. PMID: 27288264. In the supplementary data files of this paper the authors published the secretome content of young and senescent adipose mesenchymal stromal cells. Similar papers describing the proteome content of adipose mesenchymal stromal cells are available in literature.

Reviewer 2 Report

The manuscript by Rampler et al describes a very interesting work showing the integrative application of multiomics to provide a more holistic molecular perspective by integrating multiple types of omics, namely lipidomics, proteomics and metabolomics to give new insights in cell differentiation. The manuscript is well organized and technically sound but have some limitations including small sample size. Despite the low sample size, the study provide a novel approach for the proof of principle and some valuable information on the application of multiomics not only for follow-up studies but also its potential for understanding the complex pathophysiology of diseases, among other applications. However, the study suffers from other limitations that are not adequately acknowledged by the authors. The manuscript should be accepted for the publication after addressed major critique points listed below.

1. The section 4.2 Multiomics of the methods should be improved. Authors should provide all necessary experimental details namely for proteomics analysis (column, solvent system for LC-MS/MS, etc). Also, how protein extraction was performed? It was used the pellet obtained with the adapted SIMPLEX two-phase extraction preparation (as reported in page 10, line 351) or proteins were (further) extracted using guanidine chloride/HEPES buffer followed by precipitation in methanol/chloroform/water mixture as reported in page 10, line 373 and 374? The parameters of MS part should be also completed, namely the cycle structure, type of ions analyzed, etc. Why MS data used for quantification was acquired only in positive ion mode? Particularly, in the case of lipidome, negative ion mode is usually required for PI class. Moreover, the authors should clarify the composition of metabolites and lipids present in the multi mixes and all the standards used, and the range of dilutions used for external calibrations.

2. As stated above, the number of samples analyzed is very small (n=5) and in some analysis only n=3 and n=2 are used. The n=3 used for targeted analysis of lipids and metabolites (section 2.2) and the n=2 used for proteomics analysis (section 2.4) are highly small sample sizes to statistical analysis and (analytical and biological) validation. In this way, with the data available is precocious to claim on biological and analytical validation. In the case of n=2, validation is not achievable.

3. What were the 12 lipids and 54 metabolites used for targeted analysis? What was the criteria for their selection? Additionally, the MSC 21 samples were highly heterogeneous and disperse in the PC1 (44.9%) (Figure 2 A). In fact. MSC 0 and MSC 21 are only separated in PC 2, with 25.8% of the variance. Authors should provide further statistical analysis data since the PCA is not sufficient.

4. What was the sample size used for non-targeted analysis? It was a n=3 samples? Did the authors performed any normalization of the chromatographic peak areas using non-endogenous internal standards to eliminate instrumental or analytical interferences in this case of non-targeted analysis? As stated in the section 4.2 Multiomics of the methods the MS data used for quantification was acquired only in positive ion mode. In this way, how did the authors quantified the variations in PI levels? What type of ions were analyzed?

5. Did gangliosides were identified in the lipid fraction (upper phase) that corresponds to the organic layer or in the lower phase, which corresponds to the more polar layer? Gangliosides are complex and highly polar lipid compounds with relatively hydrophilic headgroups that are soluble in water and they are usually lost in the aqueous phase. Therefore, gangliosides analysis requires specialized lipid extraction methods since these lipids are not present in the organic layer where the other more simple lipids are recovered. In figure A7 A, the sphingosine d18:1-H2O (that should be assigned as [sphingosine d18:1-H2O+H]+) is not attributed to any peak in spectrum, and the ions at m/z 520.5090 should be assigned as [ceramide d18:1-H2O+H]+ since the loss of hexose is from ceramide moiety and not from the all ganglioside structure, which can lead to misunderstandings. Ions at m/z 264.2687 should be assigned as [sphingosine d18:1-2H2O+H]+. In spectrum A7 B, the ions at m/z 290.0878 should be assigned as [NeuACres-H]- and the m/z of the ions corresponding to deprotonated galactose ([galactose-H]-) should be identified. Please clarify the identification of ions at m/z 87.0087 (C3H3O3).

Specific comments.

Page 2, line 69 “appy” should be replaced by “apply”; Page 4, line 162 “< 0.05,)” should be < 0.05) without the comma (,); The “m/z” should be in italic: “m/z” (page 5 lines 218, 228, 230, 234, 235; page 6 line 239; page 9 line 295); TG (16:0_16_1_18:0) should be replaced by TG (16:0_16:1_18:0); TG (16:1_18_0_18_0) should be replaced by TG (16:1_18:0_18_0); Figure A3: add the label in the y-axis; Figure A4. Legend: The version of LipidSearch is 4.2 or 4.1 as stated in “Material and Methods” section?

Reviewer 3 Report

In the current manuscript the authors address a relatively old topic, that of integrating multiomics data, by proposing a new and original approach. The authors claim a potential for the described strategy in the context of adipogenesis, nevertheless, it can go further and reach almost any pathophysiologic question. Maybe this should be stressed by the authors.

Comments:

Globally, the results are sketched, sometimes to a minimum of information. I can understand that the biological interpretation is out of the scope of the manuscript, and that the goal was to show the workflow and validate the strategy, but some of the data are especially exciting, and would require a more in depth discussion. For example, the ganglioside finding is remarkable. Are there any hints about the reason why ganglioside concentrations are modified along adipogenesis? Did the authors find proteins involved in glycosphingolipid metabolic pathways differentially expressed? Were they identified at all, for example GM3 synthase? I suggest to better explain the procedure and the results that are represented by figure 4, so as to say, the multiomic network analysis and integration. The number of samples / determinations per experiment should be clearly stated in the figure legends. Some figures are cited in the text but I could not find them in the submission: figure S2 and B5 (line 252). Is this a typo or is it my mistake? Captions for tables should be improved. Nothing is said, for example, about the different subnetworks of Table S2. What do they mean? Also, table documents are not named as S1 and S2, as they are in the text. The multimetabolite and multilipid mixes are described in line 343, but in parenthesis it is said (x metabolites) and (x lipids). What does this mean? Is this a mistake or should the reader expect to find a precise number of molecules instead of “x”? How many proteins are identified? If I am not wrong this is not said in the text. Is it normal that in figure A4 there are no error bars?

Round 2

Reviewer 1 Report

None

Reviewer 2 Report

After going through the revised version of the manuscript, it seems that all points were successfully addressed.